# Socioeconomic Factors Contributing to Antibiotic Resistance in China: A Panel Data Analysis

**DOI:** 10.3390/antibiotics10080994

**Published:** 2021-08-17

**Authors:** Xuemei Zhen, Jingchunyu Chen, Xueshan Sun, Qiang Sun, Shasha Guo, Cecilia Stålsby Lundborg

**Affiliations:** 1Centre for Health Management and Policy Research, School of Public Health, Cheeloo College of Medicine (NHC Key Laboratory of Health Economics and Policy Research), Shandong University, Jinan 250012, China; zhenxuemei@sdu.edu.cn (X.Z.); guoshashaya@mail.sdu.edu.cn (S.G.); 2Department of Prosthodontics, School of Tomatology, Cheeloo College of Medicine, Shandong University, Jinan 250012, China; rose68042@163.com; 3Center for Health Policy Studies, School of Public Health, Zhejiang University School of Medicine, Hangzhou 310058, China; sunxueshan@zju.edu.cn; 4Department of Global Public Health, Karolinska Institutet, 171 77 Stockholm, Sweden; cecilia.stalsby.lundborg@ki.se

**Keywords:** socioeconomic factor, antibiotic resistance, hospital bed density, physician density, gross domestic product, out-of-pocket, education, China, panel data model

## Abstract

The relationship between socioeconomic factors and antibiotic resistance (ABR) prevalence remains a knowledge gap in China. In this study, our aim was to examine the association between ABR prevalence and socioeconomic factors across 30 provinces in mainland China. We used two measures of level of ABR: the proportion of methicillin-resistant *Staphylococcus aureus* (MRSA), third-generation cephalosporin-resistant *Escherichia coli* (3GCREC), and third-generation cephalosporin-resistant *Klebsiella pneumoniae* (3GCRKP), and the aggregate resistance. The data of ABR prevalence, education, gross domestic product (GDP) per capita, out-of-pocket (OOP) health expenditure, physician density, hospital bed density, and public toilet density during 2014 and 2018 in 30 provinces in mainland China were included. We examined the association between ABR prevalence and potential contributing socioeconomic factors using panel data modeling. In addition, we explored this relationship in the eastern, central, and western economic zones. Our results indicated that GDP per capita was significantly positively correlated with ABR in mainland China and the eastern economic zone; however, significantly positive associations did not exist in the central and western economic zones. Surprisingly, both higher GDP per capita and higher OOP health expenditure were associated with a higher level of MRSA, but a lower level of 3GCREC; higher physician density was associated with a lower level of MRSA, but a higher level of 3GCREC. In addition, ABR prevalence presented a decline trend during 2014 and 2018. Our study showed the potential associ-ations between resistance and GDP per capita, OOP health expenditure, physician density. It high-lights that the social and economic determinants can be of importance in tacking the development and spread of ABR in mainland China.

## 1. Introduction

Antibiotic resistance (ABR) occurs when bacteria develop the ability to defeat the antibiotics that were previously effective. ABR poses a threat to public health and hinders progress in achieving sustainable development goals (SDGs) [1]. Infections caused by ABR are associated with longer hospital stays, higher medical costs, and increased mortality compared with infections with antibiotic susceptible organisms [2,3,4,5]. It has been estimated that 100 trillion USD in economic loss and 10 million deaths every year by 2050 could be attributable to antimicrobial resistance (AMR) around the globe, and there would be 20 trillion USD in economic burden and one million deaths every year by 2050 in China due to AMR [6]. Multidrug-resistant (MDR) bacteria, as the major therapeutic challenge in both hospital and community settings [7], are leading us into a “post-antibiotic era”, in which common infections and minor injuries could be fatal [8].

The World Health Organization (WHO) developed a global priority list of MDR bacteria, in which third-generation cephalosporin-resistant *Klebsiella pneumoniae* (3GCRKP), third-generation cephalosporin-resistant *Escherichia coli* (3GCREC), and methicillin-resistant *Staphylococcus aureus* (MRSA) were regarded as the most problematic and highly MDR bacteria, as they have the capabilities for developing resistance mechanisms to a wide range of antibiotics [9]. The situation of ABR, especially the abovementioned MDR bacteria, in China is severe [10]. Reports from China Antimicrobial Resistance Surveillance System (CARSS) showed that the proportions of 3GCRKP, 3GCREC, and MRSA were 33.0%, 54.2%, and 32.2%, respectively, higher than those in some high-income countries [11].

Some studies have explored drivers of the emergence and transmission of AMR [12]. Inappropriate antibiotic use is one of the main drivers of ABR [13,14]; however, previous studies demonstrated that interventions solely focusing on regulating antibiotic consumptions had limited effectiveness [13,15,16,17,18]. The emergence and the spread of ABR are influenced by socioeconomic determinants, such as political factors, economic drivers, and sociocultural influences [15]. For example, economic status affects antibiotic prescriptions, thus further influencing ABR. Higher ABR rates and lower per capita antibiotic consumption were observed in several high- and low/middle-income countries [16,18]. Quality of governance, education, out-of-pocket (OOP) health expenditure, health facilities, and sanitation have been shown to be as closely associated with differences in ABR and antibiotic consumption patterns between regions [13,14,17,18,19,20]. Antibiotic consumptions in animals, agriculture, and aquaculture also play underappreciated roles in the development and spread of resistant bacteria [21,22].

China has vast territory that is characterized by regions with diversity in environmental, economic, and sociocultural contexts, as well as a disparity in medical resource distribution and population literacy [23]. However, the relationship between these socioeconomic factors and the level of ABR remains largely uninvestigated in China. In this study, we examined the association between provincial ABR level and socioeconomic factors across 30 provinces in mainland China.

## 2. Methods

### 2.1. Data Sources

We created a dataset on ABR for 30 provinces in mainland China using data for 5 years (2014–2018). ABR data were obtained from CARSS, showing the resistance rate with colonization or infection for each special bacterium among different provinces and autonomous regions in mainland China, which were organized by the Expert Committee on Rational Use of Drugs and National Health Commission of China. In 2018, these data covered 1353 hospitals from all provinces and autonomous regions that participated, including 349 secondary hospitals (26%) and 1004 tertiary hospitals (74%). The number of participating hospitals in 2018 among each region is presented in Figure 1 [24]. To avoid duplication, only the first episode of the same bacterium for one patient was included in each year, resulting in a total of 3,234,372 bacterial strains for analysis in 2018. According to the global priority list of MDR bacteria from WHO and the completeness and availability of resistant organisms from each province in mainland China, we included one Gram-positive bacterium (MRSA) and two Gram-negative bacteria (3GCRKP and 3GCREC) in this study. Susceptibility testing of the bacteria was determined using the disc diffusion method or automated system according to the Clinical and Laboratory Standards Institute (CLSI) criteria [24,25]. The interpretation of susceptibility testing was based on the CLSI definition and reported as resistance (R), intermediate (I), or susceptible (S) [25]. ABR combinations were aggregated by use of the average resistance prevalence of the above three bacteria. We excluded the Tibet autonomous region due to missing data [26].

Three economic zones were divided according to economic development and geographical location in China.

There are many possible contributing socio-economic factors influencing ABR prevalence; however, provincial data coverage of all factors was incomplete, therefore, we captured some main probable factors. We used the China Statistical Yearbook for data on education (finishing secondary education), gross domestic product (GDP) per capita, and public toilet density (number of public toilets per 10,000 population) between 2014 and 2018 [27]. We took data for OOP health expenditure, physician density (number of physicians per 1000 population), and hospital bed density (number of beds per 1000 population) from the China Health Statistical Yearbook during 2014 and 2018 [28].

In this study, finishing secondary education was defined as the percentage of people finishing education in junior high school in the total population aged six and over. GDP per capita was a measure of the output of a province, taking GDP and dividing it by the number of people in the province. Public toilet density, i.e., the number of public toilets per 10,000 population, was calculated as the number of public toilets divided by the total population in a province, adjusting this ratio to per 10,000 population; public toilets are rooms or small building with toilets and sinks that are available for use by the general public, customers, etc. OOP health expenditure was defined as payments made by individuals to healthcare providers for health services. Physician density, measured by the number of physicians per 1000 population, and hospital bed density, measured by the number of beds per 1000 population, were calculated as the number of physicians or number of hospital beds divided by the total population in the province, adjusting the ratios to per 1000 population.

### 2.2. Statistical Analysis

The dataset for the main analysis was panel data modeling of different bacteria at the provincial level. To explore the association between ABR prevalence and the confounding factors, we used two provincial-level measures of ABR: the proportion of MRSA, 3GCREC, and 3GCRKP and the aggregate resistance. Resistance could vary from 0–100 [18].

Fixed or random effects panel data modellings by the Hausman test were conducted to analyze the effect of confounding factors on the ABR prevalence. If the *p*-value was significant (*p* < 0.05), we used fixed effects, if not random effects. First, we estimated provincial-level regression models with aggregate resistance in 30 provinces as the outcome. In addition, we conducted a regression model in the eastern, central, and western economic zones, which were divided according to economic development and geographical location in mainland China [29] (Figure 1). Then, we ran separate regression models with MRSA, 3GCREC, and 3GCRKP groups as the dependent variables.

The independent variables in the regression analyses were selected according to our hypothesis and previous research [13,14,16,17,18,19,20]. We assumed that ABR prevalence was associated with finishing secondary education, GDP per capita, OOP health expenditure, hospital bed density, physician density, and public toilet density.

GDP per capita in 2018 USD was log-transformed to account for nonlinearity. All tests were two-tailed, and *p*-values less than 0.05 were considered significant. All statistical analyses were conducted using STATA version 14.

## 3. Results

The mean prevalence of ABR for the dependent variable aggregate resistance in 30 provinces was 41.0%, while the values for MRSA, 3GCREC, and 3GCRKP were 32.9%, 56.4%, and 33.8%, respectively. The mean aggregate resistance in eastern, central, and western economic zones was 43.0%, 43.0%, and 37.6%, respectively. The average percentage of population finishing secondary education, GDP per capita, OOP health expenditure, number of beds per 1000 population, number of physicians per 1000 population, and number of public toilets per 10,000 population in 30 provinces were 69.0%, 14,813 USD, 29.3%, 5.4, 6.3, and 2.9, respectively. Values in the eastern economic zone were 75.0%, 21,306 USD, 27.3%, 5.1, 6.8, and 2.7, respectively. Values in the central economic zone were 70.6%, 12,305 USD, 32.9%, 5.5, 5.7, and 2.9, respectively. Values in the western economic zone were 61.8%, 11,787 USD, 28.8%, 5.8, 6.3, and 3.3, respectively (Table 1).

Table 2 shows estimates from regression with percent aggregate resistance in 30 provinces and in three different economic zones as the dependent variable. First, we only noted a significantly positive correlation between ABR prevalence and GDP per capita (95% confidence interval (CI): 0.22–23.07, *p* = 0.046). In addition, we also found a positive relationship between GDP per capita and ABR prevalence in the eastern economic zone; however, there were no significant differences in the central and western economic zones. Moreover, time was an important factor, with downward trends for ABR prevalence found in 30 provinces, as well as in the eastern, central, and western economic zones (Table 2).

Table 3 shows estimates from the regression with the percentage of resistant isolates for each separate bacterium as the outcome variable. We explored a statistically significantly positive relationship between GDP per capita and MRSA prevalence and between physician density and 3GCREC prevalence, but a statistically significantly negative relationship between OOP health expenditure and 3GCRKP prevalence and between physician density and MRSA prevalence. In addition, OOP health expenditure was marginally significantly positively associated with MRSA prevalence, whereas a marginally significantly negative association was identified between GDP per capita and 3GCREC prevalence. Time was also an important factor underlying MRSA, 3GCREC, and 3GCRKP prevalence (Table 3).

## 4. Discussion

Our results indicated that GDP per capita was significantly positively correlated with ABR in mainland China and the eastern economic zone; however, significantly positive associations did not exist in the central and western economic zones. Surprisingly, higher GDP per capita and higher OOP health expenditure were associated with a higher level of MRSA, but a lower level of 3GCREC; higher physician density was associated with a lower level of MRSA, but a higher level of 3GCREC. In addition, ABR prevalence exhibited a declining trend during 2014 and 2018.

A series of national efforts were announced by the Chinese Government to combat ABR, which produced good results in terms of a downward trend of MRSA, 3GCREC, and 3GCRKP prevalence. From 2011 to 2013, a national campaign in healthcare institutions was run to promote the rational use of antibiotics, mainly in secondary and tertiary hospitals [30]. In 2015, the use of four antibiotics in animals, namely, lomefloxacine, pefloxacine, ofloxacin, and norfloxacin, was banned by the Ministry of Agriculture and Rural Affairs [31]. In 2016, China initiated a 5 year national action plan to curb the increasing use of antibiotics and ABR from 2016 to 2020 in response to the WHO global efforts [30]. In addition, China has collaborated with the UK, Sweden, and European Union in efforts to combat ABR over the past decade.

One study reported that a negative and significant correlation between GDP per capita and ABR prevalence existed at the global level [16]. Poverty itself might not be associ-ated with a direct effect on ABR level, but this would be influenced by factors such as sanitation, medical environmental. Countries with limited resources suffer from inappropriate use of antibiotics with inadequate treatment duration and dosing, medication sharing, or overuse of medication [16]. In addition, poor environmental sanitation, inappropriate antibiotic usage, low vaccination rates, and poor laboratory and infection control capacity in resource-poor regions might lead to the prescription of last-resort antibiotics while ignoring the problem of ABR [16,19]. However, there were no previous studies exploring the association between GDP per capita and ABR prevalence within countries. In our study, we found that GDP per capita was significantly positively associated with the level of ABR in 30 provinces and in the eastern economic zone. It was reported that countries with high GDP per capita were known to have high antibiotic consumption rates, which might be contributing more to the burden of ABR; [13,32]. Opposite results regarding the association between GDP per capita and ABR prevalence might be related to differences in resistance patterns and antimicrobial stewardship programs. China has a vast territory, whereby socioeconomic development and medical resources vary widely and antibiotic-prescribing habits and resistance patterns are of great variety [29]. This suggests that provinces should adopt antibiotic stewardship for ABR according to local condition.

There were two opposite results in terms of socioeconomic factors contrasting MRSA and 3GCREC/3GCRKP, which might be related to bacterial types and resistant patterns. Patients with a higher percentage of OOP health expenditure might consume inexpensive and narrow-spectrum antibiotics, which might result in a lower degree of 3GCRKP. In addition, supplier-induced demand was an important determinant factor for excess use of healthcare [33,34]. Less control of oxacillin or cefoxitin, as well as the length of time of drug therapy and dosing used, might have resulted in the significant and positive correlation between OOP health expenditure and MRSA level [13,14,17]. A positive link was identified between antibiotic usage and ABR [35] and between OOP health expenditure and antibiotic consumption in European countries [17]. In general, physicians played an important role in hospitals that increased ABR level by over-prescribing antibiotics, especially expensive and broad-spectrum antibiotics, which might put forward some evidence of supply-induced demand [36]. In addition, physicians also played a positive role in patient adherence to treatment and lower patient contribution to ABR, further exacerbating the problem of ABR [37].

There is no comprehensive, universally agreed definition of governance [38], and practices defined as governance might vary depending on country and context [39]. Because of data gaps, the indicator of governance was difficult to measure; hence, we did not include it. In addition, the focus was mainly on antibiotic consumption as the most important factor contributing to ABR in previous studies [14,18]. Although we did not assess the association between antibiotic consumption and ABR in this study, because adequate data were lacking, the findings in other studies confirmed that many social and economic factors are of more importance than antibiotic consumption in explaining the diversity in the level of ABR in different region [14,18].

To our best knowledge, this is the first cross-regional examination of the societal economic determinants of the ABR level using panel data modeling within countries, including China. It is also the first study to compare the predictors of ABR in three different economic zones and between Gram-positive organisms and Gram-negative organisms in mainland China. However, there were some limitations in our study. Firstly, some confounding factors in the human and agricultural sectors, as well as the environment, that might correlate with ABR level could not be collected from each province, such as temperature, antibiotic consumption, and governance, due to inadequate data. Relevant studies including the above socioeconomic factors using real-world data are needed in the future. Secondly, a relationship between ABR level and socioeconomic factors was presented; however, there was no definitive evidence of ABR mechanism due to limited bacteria and risk factors. In addition, we acknowledge the ABR prevalence data from CARSS as the best approach to represent the provincial level; however, this could not be applied to primary care facilities, and any conclusions might not be extrapolated to other countries. Lastly, due to the time limitation, we did not explore changes over time in terms of the prevalence of resistance and the socio demographic data, which will be conducted in the future using CHINET data to show the changing trends of bacterial resistance since 2005 in China [24].

## 5. Conclusions

Our results found that GDP per capita was significantly positively correlated with ABR in mainland China and the eastern economic zone. However, there were two opposite results in terms of GDP per capita, OOP health expenditure, and physician density contrasting MRSA and 3GCREC/3GCRKP, which might depend on bacterial types and resistant patterns. In addition, ABR prevalence exhibited a decline trend during 2014 and 2018. Our study highlights that intervention measures tackling the development and spread of ABR in mainland China must better recognize and address the importance of social and economic determinants. We also need to focus on bacterial types and resistant patterns. Despite China having made great efforts to reduce the level of ABR in recent years, antimicrobial stewardship needs to be further strengthened.

## Figures and Tables

**Figure 1 antibiotics-10-00994-f001:**
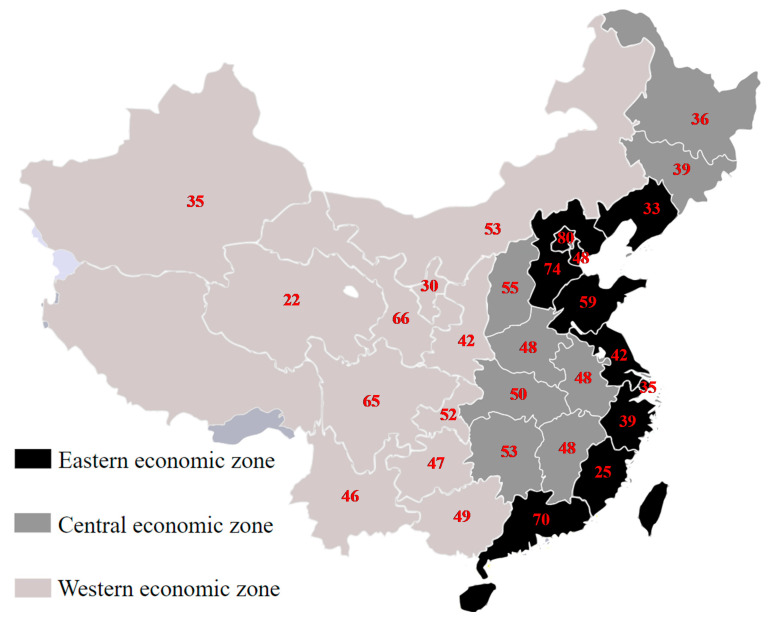
Three economic zones in mainland China (30 provinces). The number with red font indicates the number of participating hospitals in 2018 in the China Antimicrobial Resistance Surveillance System (CARSS) in each province and autonomous region. We excluded the Tibet autonomous region due to missing data.

**Table 1 antibiotics-10-00994-t001:** Variables and measures.

Variables	30 Provinces		Eastern Economic Zone		Central Economic Zone		Western Economic Zone	
	Mean	SD	Mean	SD	Mean	SD	Mean	SD
Aggregate resistance, %	41.0	6.7	43.0	6.4	43.0	7.9	37.6	4.6
MRSA, %	32.9	8.4	35.6	8.5	32.9	9.2	30.2	6.9
3GCREC, %	56.4	5.4	56.5	5.2	58.7	6.0	54.6	4.5
3GCRKP, %	33.8	9.5	37.0	7.8	37.3	10.2	28.0	7.7
Education, % of finishing secondary education	69.0	8.6	75.0	7.6	70.6	3.9	61.8	6.7
GDP per capita, log	14,813.5	1.5	21,306.2	1.5	12,304.8	1.2	11,787.3	1.4
OOP health expenditure, % of total health expenditures	29.3	4.9	27.3	5.8	32.9	3.4	28.8	3.3
Hospital bed density, number of beds per 1000 population	5.4	0.8	5.1	0.8	5.5	0.7	5.8	0.7
Physician density, number of physicians per 1000 population	6.3	1.2	6.8	1.5	5.7	0.7	6.3	0.9
Public toilet density, number of public toilets per 10,000 population	2.9	1.1	2.7	0.8	2.9	0.8	3.3	1.4

MRSA: methicillin-resistant *Staphylococcus aureus*; 3GCREC: third-generation cephalosporin-resistant *Escherichia coli*; 3GCRKP: third-generation cephalosporin-resistant *Klebsiella pneumoniae*; SD: standard deviation; CI: confidence interval; GDP gross domestic product; OOP: out-of-pocket.

**Table 2 antibiotics-10-00994-t002:** Effect of variables on the aggregate resistance (including MRSA, 3GCREC, and 3GCRKP) among 30 provinces, as well as in the eastern economic zone, central economic zone, and western economic zone in mainland China (excluding the Tibet autonomous region).

Variables	30 Provinces					Eastern Economic Zone					Central Economic Zone					Western Economic Zone				
	B	SE	*p*-Value	95% CI		B	SE	*p*-Value	95% CI		B	SE	*p*-Value	95% CI		B	SE	*p*-Value	95% CI	
Education, % of finishing secondary education	0.08	0.13	0.547	−0.17	0.33	−0.13	0.24	0.581	−0.62	0.35	−0.37	0.36	0.303	−1.07	0.33	0.14	0.15	0.361	−0.16	0.43
GDP per capita, log	11.64	5.76	0.046	0.22	23.07	26.64	8.54	0.004	9.28	44.00	28.24	24.31	0.245	−19.40	75.89	3.40	10.11	0.739	−17.15	23.95
OOP health expenditure, % of total health expenditures	−0.09	0.12	0.458	−0.33	0.15	−0.01	0.28	0.967	−0.57	0.55	−0.70	0.45	0.116	−1.58	0.17	−0.05	0.17	0.746	−0.39	0.28
Hospital bed density, number of beds per 1000 population	−0.87	0.90	0.339	−2.66	0.92	−0.08	1.93	0.966	−4.00	3.84	−2.74	2.79	0.326	−8.21	2.73	1.21	1.24	0.335	−1.30	3.72
Physician density, number of physicians per 1000 population	0.18	0.87	0.836	−1.54	1.90	−0.66	1.80	0.717	−4.32	3.00	−0.05	3.41	0.989	−6.74	6.64	−1.85	1.11	0.106	−4.11	0.41
Public toilet density, number of public toilets per 10,000 population	0.04	0.26	0.889	−0.48	0.55	−0.51	0.41	0.221	−1.33	0.32	1.05	1.24	0.397	−1.38	3.49	−0.04	0.38	0.923	−0.81	0.74
Year																				
2015	−2.66	1.11	0.018	−4.86	-0.46	−5.52	1.61	0.002	−8.78	−2.25	−6.79	5.20	0.192	−16.99	3.40	−0.87	1.73	0.617	−4.39	2.64
2016	−3.52	0.81	<0.000	−5.12	-1.92	−3.68	1.23	0.005	−6.18	−1.18	−6.37	3.31	0.055	−12.86	0.13	−2.85	0.99	0.007	−4.86	−0.83
2017	−6.73	1.42	<0.000	−9.55	-3.91	−8.36	2.32	0.001	−13.07	−3.64	−11.87	6.24	0.057	−24.10	0.36	−4.36	1.79	0.020	−8.00	−0.73
2018	−6.77	1.45	<0.000	−9.65	-3.89	−6.59	2.49	0.012	−11.64	−1.53	−9.86	5.18	0.057	−20.00	0.29	−5.22	1.87	0.009	−9.03	−1.42
(Constant)	0.08	0.13	0.547	−0.17	0.33	−0.13	0.24	0.581	−0.62	0.35	-0.37	0.36	0.303	−1.07	0.33	0.14	0.15	0.361	−0.16	0.43
*p*-value	<0.000					<0.000					<0.000					<0.000				
*R* ^2^	0.768					0.817					0.823					0.828				
Model (Hausman test)	Fixed effects (0.8969)					Fixed effects (0.0927)					Random effects (0.0441)					Fixed effects (0.8549)				

MRSA: methicillin-resistant *Staphylococcus aureus*; 3GCREC: third-generation cephalosporin-resistant *Escherichia coli*; 3GCRKP: third-generation cephalosporin-resistant *Klebsiella pneumoniae*; SE: standard error; CI: confidence interval; GDP gross domestic product; OOP: out-of-pocket; B: regression coefficient; *R*^2^: coefficient of determination.

**Table 3 antibiotics-10-00994-t003:** Effect of variables on the MRSA, 3GCREC, and 3GCRKP prevalence among 30 provinces in mainland China (excluding the Tibet autonomous region).

Variables	MRSA					3GCREC					3GCRKP				
	B	SE	*p*-Value	95% CI		B	SE	*p*-Value	95% CI		B	SE	*p*-Value	95% CI	
Education, % of finishing secondary education	−0.03	0.22	0.887	−0.46	0.40	0.03	0.13	0.788	−0.22	0.29	0.22	0.21	0.276	−0.18	0.63
GDP per capita, log	44.49	9.95	<0.000	24.77	64.20	−11.43	5.86	0.054	−23.05	0.19	1.84	9.41	0.845	−16.81	20.49
OOP health expenditure, % of total health expenditures	0.38	0.21	0.070	−0.03	0.80	−0.04	0.12	0.750	−0.28	0.20	−0.61	0.20	0.002	−1.01	−0.22
Hospital bed density, number of beds per 1000 population	0.52	1.56	0.740	−2.57	3.61	−1.22	0.92	0.187	−3.04	0.60	−1.91	1.48	0.198	−4.84	1.01
Physician density, number of physicians per 1000 population	−3.04	1.50	0.045	−6.01	−0.07	2.10	0.88	0.019	0.35	3.85	1.48	1.42	0.298	−1.33	4.29
Public toilet density, number of public toilets per 10,000 population	0.60	0.45	0.183	−0.29	1.48	−0.24	0.26	0.361	−0.76	0.28	−0.25	0.42	0.557	−1.09	0.59
Year															
2015	−6.75	1.91	0.001	−10.54	−2.96	0.90	1.13	0.428	−1.34	3.13	−2.11	1.81	0.245	−5.70	1.47
2016	−1.96	1.39	0.161	−4.72	0.80	−3.29	0.82	<0.000	−4.91	−1.66	−5.31	1.32	<0.000	−7.92	−2.70
2017	−8.79	2.45	0.001	−13.65	−3.92	−4.41	1.45	0.003	−7.28	−1.54	−6.99	2.32	0.003	−11.59	−2.39
2018	−5.45	2.51	0.032	−10.41	−0.48	−7.01	1.48	<0.000	−9.94	−4.09	−7.84	2.37	0.001	−12.54	−3.14
(Constant)	−0.03	0.22	0.887	−0.46	0.40	0.03	0.13	0.788	−0.22	0.29	0.22	0.21	0.276	−0.18	0.63
*p*-value	<0.000					<0.000					<0.000				
*R* ^2^	0.580					0.791					0.527				
Model (Hausman test)	Fixed-effects (0.2943)					Fixed-effects (0.5773)					Fixed-effects (0.4293)				

MRSA: methicillin-resistant *Staphylococcus aureus*; 3GCREC: third-generation cephalosporin-resistant *Escherichia coli*; 3GCRKP: third-generation cephalosporin-resistant *Klebsiella pneumoniae*; SE: standard error; CI: confidence interval; GDP: gross domestic product; OOP: out-of-pocket; B: regression coefficient; *R*^2^: coefficient of determination.

## Data Availability

All data generated or analyzed during this study are included in the tables and figures.

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
