# Peer review of "Socioeconomic Factors Contributing to Antibiotic Resistance in China: A Panel Data Analysis"

_antibiotics, 2021, doi:10.3390/antibiotics10080994_

Round 1
Reviewer 1 Report
The authors performed panel data analysis to identify social and economic determinants to development and spread of antibiotic resistance in China.
In general, the paper is well written. However, there are many instances where sentence structure and/or typos detract from the overall readability of the manuscript.
Major comments
-. In Table 2, fixed effects and random effects were used in different models. Authors should discuss how different effects were chosen for different models. Also, please describe what variable(s) was modeled for random effects.
-. Reasoning or interpretation of why authors has obtained the opposite result about the association between GDP per capita and ABR prevalence compared to the other studies is missing.
Minor comments
-. ABR is used for the first time in Abstract.
-. ‘Year’ might not be necessary in Table2.
-. Bacterial and resistant patterns in multiple places -> bacterial resistant patterns
Author Response
The authors performed panel data analysis to identify social and economic determinants to development and spread of antibiotic resistance in China.
In general, the paper is well written. However, there are many instances where sentence structure and/or typos detract from the overall readability of the manuscript.
Major comments:
(1) In Table 2, fixed effects and random effects were used in different models. Authors should discuss how different effects were chosen for different models. Also, please describe what variable(s) was modeled for random effects.
Response: Fixed or random effects were used according to hausman test where P-value>0.05 is that the preferred model is fixed effects. We choose the same variables including education, GDP per capita, OOP health expenditure, hospital bed density, physician density, public toilet density, year for random effects and fixed effects.
Please see in the Methods-Statistical analysis section, lines 112-113, page 6; Table 2 and Table 3.
(2) Reasoning or interpretation of why authors has obtained the opposite result about the association between GDP per capita and ABR prevalence compared to the other studies is missing.
Response: We have explained the reason.
The opposite results about the association between GDP per capita and ABR prevalence might be related to differences in resistance patterns and antimicrobial stewardship programs. China has vast territory, where socioeconomic development and medical resources vary widely and antibiotic prescribing habits and resistant patterns are of great variety.
Please see in the Discussion section, lines 177-180, page 11.
Minor comments
(3) ABR is used for the first time in Abstract.
Response: We are so sorry for our careless. We have defined the ABR for the first time in abstract.
Please see in the Abstract section, line 23, page 2.
(4) ‘Year’ might not be necessary in Table2.
Response: Because we created a dataset on ABR for 30 provinces in mainland China using data for five years (2014-2018), all data over 5 year period put together is confusing, so we conducted panel data modelling with time, and found “year” is an important factor of ABR level.
Please see in Methods-Data sources section, line 73, page 4; Results section, lines 147-148, page 10; Results section, lines 154-155, page 10.
(4) Bacterial and resistant patterns in multiple places -> bacterial resistant patterns
Response: Thank you very much for your comments, we have revised it to bacterial types and resistant patterns.
Please see in Discussion section, line 184, page 11; Conclusion section, line 218, line 221, page 13.
Reviewer 2 Report
This is an important study and excellently written manuscript. The references are up to date and authors have included all the important aspects of microbial resistance in the introduction. However, I have only few suggestions for improvement:
The sentence: “ABR data were obtained from CARSS, which is an operational national surveillance network for bacterial resistance with well-known in China.” I do not understand the need for the word with in this sentence.
The sentence: “Susceptibility testing of the bacteria was determined using the disk diffusion method or automated system according to the Clinical and Laboratory Standards Institute (CLSI) criteria.” Please provide readers with exact protocol, equipment and chemicals used for this testing as it would help with reproducibility of your study in the future. Furthermore, provide reference for this testing.
Table 1 - all the other numbers are written with one decimal place so add it to GDP as well
Author Response
This is an important study and excellently written manuscript. The references are up to date and authors have included all the important aspects of microbial resistance in the introduction. However, I have only few suggestions for improvement:
(1) The sentence: “ABR data were obtained from CARSS, which is an operational national surveillance network for bacterial resistance with well-known in China.” I do not understand the need for the word with in this sentence.
Response: Thank you very much for your comments. We agreed that this sentence is not necessary, so we delete it.
Please see in Methods-Data sources section, lines 73-76, page 4.
(2) The sentence: “Susceptibility testing of the bacteria was determined using the disk diffusion method or automated system according to the Clinical and Laboratory Standards Institute (CLSI) criteria.” Please provide readers with exact protocol, equipment and chemicals used for this testing as it would help with reproducibility of your study in the future. Furthermore, provide reference for this testing.
Response: The details about susceptibility testing were reported in some published studies, we have added the references.
Please see in Methods-Data sources section, line 84, page 4.
Table 1 - all the other numbers are written with one decimal place so add it to GDP as well
Response: We have revised it accordingly.
Please see in Table 1.
Reviewer 3 Report
Dear authors,
I suggest you
-to delete the last sentences from the abstract from "The need to .... "(they are repeated in the conclusion and not useful for the reader),
-to write in the first row of the abstract Antibiotic resistance (ABR).
The sentence "ABR data were obtained from CARSS, which is an operational national surveillance network for bacterial resistance with well-known in China" seems incomplete.
Please rephrase the sentence "There were two opposite results in terms of socioeconomic factors contributing to resistance isolates from gram-positive organism and gram-negative organism, which might be related to bacterial and resistant patterns" in the results.
In the sentence "To our best of knowledge, it is the first cross-regional examination of the societal economic determinants of level of ABR using panel data modelling within countries, including China[15]. It is also the first study to compare the predictors of ABR in three different economic zones and between gram-positive organism and gram-negative organism in mainland China." it is not clear if you talk about your results or those from the ref 15.
What do you mean with the sentence "Second, it was presented with a relationship between ABR level and socio-economic factors, but not a definitive evidence of ABR mechanism."?
Please revise the sentence "to resistance isolates from gram-positive organism and gram-negative organism, which might depend on bacterial and resistant pattern." in the conclusion.
Author Response
I suggest you
(1) to delete the last sentences from the abstract from "The need to .... "(they are repeated in the conclusion and not useful for the reader),
Response: We have deleted it accordingly.
Please see in Abstract section, lines 38-39, page 2.
(2)to write in the first row of the abstract Antibiotic resistance (ABR).
Response: We are so sorry for our careless. We have defined the ABR for the first time in abstract.
Please see in the Abstract section, line 23, page 2.
(3) The sentence "ABR data were obtained from CARSS, which is an operational national surveillance network for bacterial resistance with well-known in China" seems incomplete.
Response: We have revised this sentence.
ABR data were obtained from CARSS, which showed the resistance rate with colonization or infection for each special bacterium among different provinces and autonomous regions in mainland China, and organized by the Expert Committee on Rational Use of Drugs and National Health Commission of China.
Please see in Methods-Data sources section, lines 73-76, page 4.
(4) Please rephrase the sentence "There were two opposite results in terms of socioeconomic factors contributing to resistance isolates from gram-positive organism and gram-negative organism, which might be related to bacterial and resistant patterns" in the results.
Response: We have rephrased this sentence.
There were two opposite results in terms of socioeconomic factors between MRSA and 3GCREC, 3GCRKP, which might be related to bacterial type and resistant patterns.
Please see in Discussion section, lines 183-184, page 11.
(5) In the sentence "To our best of knowledge, it is the first cross-regional examination of the societal economic determinants of level of ABR using panel data modelling within countries, including China[15]. It is also the first study to compare the predictors of ABR in three different economic zones and between gram-positive organism and gram-negative organism in mainland China." it is not clear if you talk about your results or those from the ref 15.
Response: We would like to discussion the results in our study, so we delete the reference.
Please see in the Discussion section, line 202, page 12.
(6) What do you mean with the sentence "Second, it was presented with a relationship between ABR level and socio-economic factors, but not a definitive evidence of ABR mechanism."?
Response: We have revised this sentence.
It was presented with a relationship between ABR level and socio-economic factors, but not a definitive evidence of ABR mechanism due to limited bacteria and risk factors.
Please see in Discussion section, lines 207-208, page 12.
(7) Please revise the sentence "to resistance isolates from gram-positive organism and gram-negative organism, which might depend on bacterial and resistant pattern." in the conclusion.
Response: We have revised this sentence.
However, there were two opposite results in terms of GDP per capita, OOP health expenditure, and physician density between MRSA and 3GCREC, 3GCRKP, which might depend on bacterial types and resistant patterns.
Please see in Conclusion section, lines 217-218, page 13.